# Salinity Treatments Promote the Accumulations of Momilactones and Phenolic Compounds in Germinated Brown Rice

**DOI:** 10.3390/foods12132501

**Published:** 2023-06-27

**Authors:** Mehedi Hasan, Nguyen Van Quan, La Hoang Anh, Tran Dang Khanh, Tran Dang Xuan

**Affiliations:** 1Graduate School of Advanced Science and Engineering, Hiroshima University, 1-5-1 Kagamiyama, Higashi-Hiroshima 739-8529, Japan; mehedihasanjony110@gmail.com (M.H.); nvquan@hiroshima-u.ac.jp (N.V.Q.); hoanganh6920@gmail.com (L.H.A.); 2Center for the Planetary Health and Innovation Science (PHIS), The IDEC Institute, Hiroshima University, 1-5-1 Kagamiyama, Higashi-Hiroshima 739-8529, Japan; 3Agricultural Genetics Institute, Pham Van Dong Street, Hanoi 122000, Vietnam; 4Center for Agricultural Innovation, Vietnam National University of Agriculture, Hanoi 131000, Vietnam

**Keywords:** momilactones, phenolics, antioxidants, bioactive compounds, germinated brown rice, salinity

## Abstract

This is the first investigation, conducted in a completely randomized design (CRD), to determine the effects of different salinity levels (75 and 150 mM) and germination periods (3, 4, and 5 days) on momilactone and phenolic accumulations in germinated brown rice (GBR) var. Koshihikari. Particularly, the identification of bioactive compounds was confirmed using electrospray ionization-mass spectrometry (ESI-MS) and nuclear magnetic resonance (NMR) spectroscopy (^1^H and ^13^C). Momilactone A (MA) and momilactone B (MB) amounts were determined by ultra-performance liquid chromatography–electrospray ionization-mass spectrometry (UPLC–ESI-MS), whereas other compounds were quantified by spectrophotometry and high-performance liquid chromatography (HPLC). Accordingly, GBR under B2 treatment (75 mM salinity for 4 days) showed the greatest total phenolic and flavonoid contents (14.50 mg gallic acid and 11.06 mg rutin equivalents, respectively, per g dry weight). GBR treated with B2 also accumulated the highest quantities of MA, MB, *ρ*-coumaric, ferulic, cinnamic, salicylic acids, and tricin (18.94, 41.00, 93.77, 139.03, 46.05, 596.26, and 107.63 µg/g DW, respectively), which were consistent with the strongest antiradical activities in DPPH and ABTS assays (IC_50_ = 1.58 and 1.78 mg/mL, respectively). These findings have implications for promoting the value of GBR consumption and rice-based products that benefit human health.

## 1. Introduction

Rice (*Oryza sativa* L.) provides about 20% of the world’s dietary energy, which is comparatively higher than wheat (19%) and maize (5%) [1]. Rice contains various secondary metabolites, including phenolic acids, flavonoids, terpenoids, steroids, and alkaloids [2]. Nowadays, rice not only plays a vital role as an indispensable food source but has also been demonstrated to possess certain health benefits for human consumption. Notably, despite brown rice (BR) possessing a higher nutritional and bioactive composition in its bran and embryo, it is less popular than white rice (WR) [1,3]. BR includes around 2% of the total dietary fiber and serves as a vital source of γ-oryzanol, vitamin E, minerals, phenolic compounds, phytosterols, and phytic acid. Therefore, the utilization of brown rice as a nutritional and functional food has emerged as a recent trend. However, due to the compact structure of its outer bran layer, brown rice tends to have a firmer texture, making it more challenging to process and less digestible compared to white rice.

As an inevitable consequence, germinated brown rice (GBR) has been found to have effective alternative features while still maintaining its inherent nutritional value. The quality of GBR is enhanced through increased water absorption on the outer kernel, resulting in a softened texture. Additionally, enzymatic activities during seed germination modify the bioactive substances through interactions between proteins and carbohydrates in the grain endosperm [4,5]. Accordingly, GBR has been reported to have a proliferation of bio-functional constituents such as γ-aminobutyric acid (GABA) [1,6], vitamins, and amino acids [7], as well as a reduction of sugar [8], compared to non-GBR. On the other hand, previous studies indicated that subjecting germinated brown rice (GBR) to abiotic stresses and various germination conditions can lead to improved nutritional profiles and an elevated accumulation of bioactive compounds and antioxidant properties. Different soaking and germination periods revealed stimulatory effects on the growth of sprouts and increased contents of total phenolics, total flavonoids, and GABA in GBR [9,10,11]. Meanwhile, abiotic stresses such as salt and cold conditions may improve the contents of GABA, polyphenols, and antioxidant activity [6,12]. Therefore, the utilization of abiotic stresses and diverse germination conditions for GBR presents a promising strategy to promote the consumption value of brown rice. In a recent study, Choe et al. [12] reported that GBR accumulated a greater content of polyphenols and flavonoids, which was in line with the motivated antioxidant activity during the treatments with calcium chloride (CaCl_2_). Nevertheless, none of the researchers focused on the effects of salinity treatment on the accumulation of bioactive compounds and antioxidant capacity of GBR.

In rice, although present in relatively small quantities, secondary metabolites such as phenolics, terpenes, and lactones play significant roles in both nutritional value and physiological processes, including metabolism, synthesis, and responses to environmental factors. For example, tricin, an important flavonoid, can be isolated from various rice plant organs (grains, leaves, brans, and husks). Tricin has been reported to have potentials for antioxidants [13], anti-skin aging [13], and anticancer [14,15,16] in numerous studies. Additionally, in rice, more prevalent are phenolic acids such as *ρ*-coumaric, ferulic, cinnamic, and salicylic acids, which have been recognized for their bioactive properties, including antioxidant, anti-inflammatory, and anticancer activities [17]. Notably, these phenolic and flavonoid compounds are accumulated with dominant contents in the bran layer [3], so they are generally found in greater amounts in BR compared to WR [18,19]. Moreover, the quantities of these phenolic compounds in GBR are up to twice higher than those in BR [3]. On the other hand, momilactones A (MA) and B (MB) have been acknowledged as valuable diterpene lactones derived from rice, which have recently exhibited antioxidant [13], anticancer (leukemia [20], lymphoma [21], and colon cancer [22]), anti-diabetes [23,24], anti-obesity [24], and anti-skin aging [13] properties. Though MA and MB have shown high potential for medicinal and pharmaceutical purposes, their contents in GBR have never been elucidated [23]. Alongside the mentioned valuable compounds, antioxidant property is also an integral criterion determining the value of rice consumption [25]. Notably, 2,2-diphenyl-1-1-picrylhydrazyl (DPPH) and 2,2′-azino-bis(3-ethylbenzothiazoline-6-sulfonic acid) (ABTS) assays are the most popular and convenient protocols to examine the antioxidant capacity of rice samples [26]. Of which, the ABTS assay relies on the production of a blue/green ABTS^•+^ radical cation, while the DPPH assay involves the reduction of the purple-colored DPPH^•^ radical to 1,1-diphenyl-2-picryl hydrazine [26]. In addition, ABTS and DPPH radicals also exhibit differences in molecular weight, stability, affinity, solubility, absorption ability, and pH requirements [26,27]. Generally, both DPPH and ABTS assays have their advantages and limitations, which should be applied in combination to obtain a more comprehensive understanding of the antioxidant properties of target products.

Among rice varieties, Koshihikari is a famous Japonica model rice cultivar that is widely distributed throughout Japan [28]. Koshihikari rice grains are small, plump, relatively lightweight, and have a rounded shape [28]. They exhibit a light brown or tan color and a smooth, glossy texture [28]. Meanwhile, the rice husks are typically thin, light, and have a pale brown color [28]. Owing to its favorable physical attributes, accompanied by its well-established aroma and taste, Koshihikari rice has garnered extensive popularity and preference among consumers [28]. However, the predominant cultivation of this particular variety contributes annually to a substantial production of rice by-products, including brans and husks, which have historically been subjected to inadequate utilization or wastage [13,24]. Conversely, scientific investigations have revealed the presence of valuable bioactive compounds within these by-products, exhibiting significant health-promoting effects [13,23,24]. Accordingly, the objective of this research endeavor was to procure Koshihikari rice husks for the purpose of isolating bioactive compounds, with a specific focus on phenolics and momilactones. Moreover, the study aimed to assess the variations in these compounds within Koshihikari BR seeds that were subjected to different salt treatments (0, 75, and 150 mM) and varying durations of germination (3, 4, and 5 days). Furthermore, an investigation was conducted to explore the correlation between the levels of bioactive compounds and the antioxidant capacity exhibited during exposure to salt conditions.

## 2. Materials and Methods

### 2.1. Materials

Rice (*Oryza sativa* var. Koshihikari) husks were collected from rice mills allocated near Hiroshima University, Higashi-Hiroshima Campus, Japan, in September 2019. In the specification, mature and healthy rice grains were selected for milling to ensure the quality of the husks. After that, the obtained husks were thoroughly cleaned with water to remove dust and impurities. The husk samples were then dried and preserved (voucher specimen: KOS-MOMI 19HJ) at the laboratory of Plant Physiology and Biochemistry, Graduate School of Advanced Science and Engineering, Hiroshima University, Japan. Brown rice of the Koshihikari was purchased from a Japan Agriculture (JA) shop in Hiroshima, Japan, to prepare germinated brown rice (GBR).

For extraction and isolation processes, methanol, hexane, and ethyl acetate were purchased from Junsei Chemical Co., Ltd. (Tokyo, Japan), while silica gel was bought from Sigma-Aldrich (St. Louis, MO, USA). Isolated compounds were dissolved in deuterated dimethyl sulfoxide (DMSO-d_6_) and deuterated chloroform (CDCl_3_) procured from Sigma-Aldrich (St. Louis, MO, USA). Standards comprising ferulic acid, cinnamic acid, and salicylic acid and chemicals including sodium acetate (CH_3_COONa), sodium carbonate (Na_2_CO_3_), sodium hypochlorite (NaClO), aluminum chloride (AlCl_3_), Folin–Ciocalteu’s reagent, 2,2-diphenyl-1-picrylhydrazyl (DPPH), potassium persulfate (K_2_S_2_O_8_), and 2,2′-azinobis(3-ethylbenzothiazoline-6-sulfonic acid) (ABTS) were acquired from Kanto Chemical Co., Inc. (Tokyo, Japan). Formic acid, trifluoroacetic acid, acetonitrile, methanol plus, and distilled water used for HPLC and UPLC analyses were obtained from Sigma-Aldrich (St. Louis, MO, USA), EMD Millipore Corporation (Billerica, MA, USA), Fisher Chemical (Hampton, VA, USA), Kanto Chemical Co., Inc. (Tokyo, Japan), and Nacalai Tesque (Kyoto, Japan), respectively.

### 2.2. Isolation of Tricin, ρ-Coumaric Acid, and Momilactones A (MA) and B (MB)

The isolation process for tricin, *ρ*-coumaric acid, and momilactones A (MA) and B (MB) followed the methods described in the previous study [13]. Briefly, 30 kg of rice husks were dried in an oven at 50 °C for six days and then extracted with 100% MeOH for two weeks at room temperature. The MeOH crude extract was then mixed with an appropriate amount of distilled water and partitioned sequentially with hexane and EtOAc. Next, the obtained EtOAc extract was subjected to column chromatography using silica gel as the stationary phase and a hexane:EtOAc (*v*/*v*) mixture as the mobile phase. MA and MB were isolated from the eluate of hexane:EtOAc (8:2, *v*/*v*), while tricin and *ρ*-coumaric acid were purified from the eluate of hexane:EtOAc (7:3, *v*/*v*).

### 2.3. Confirmation of Isolated Tricin, ρ-Coumaric Acid, and Momilactones A (MA) and B (MB) by ^1^H- and ^13^C-Nuclear Magnetic Resonance (NMR) and Electrospray Ionization-Mass Spectrometry (ESI-MS)

The identification of isolated tricin, *ρ*-coumaric acid, MA, and MB was confirmed by ^1^H- and ^13^C-nuclear magnetic resonance (NMR) spectra. Of which, ^1^H- and ^13^C NMR spectra of *ρ*-coumaric acid (in DMSO-d_6_) were received on an NMR spectrometer (Bruker Ascend 400, BRUKER BioSpin, Fällanden, Switzerland) at 400 and 101 MHz, respectively. Meanwhile, ^1^H- and ^13^C NMR spectra of tricin (in DMSO-d_6_) and MA and MB (in CDCl_3_) were acquired on an NMR spectrometer (JNM-ECA600, JEOL Ltd., Tokyo, Japan) at 600 and 151 MHz, respectively. Coupling constants (J) and chemical shifts (δ) were indicated in Hz and parts per million (ppm), respectively. The shorthand notations s, d, t, q, dd, and dt represent the resonance multiplicities singlet, doublet, triplet, quartet, doublet of doublets, and doublet of triplets, respectively.

Additionally, tricin, *ρ*-coumaric acid, MA, and MB were confirmed using electrospray ionization-mass spectrometry (ESI-MS) (LTQ Orbitrap XL, Thermo Fisher Scientific, Waltham, MA, USA). The compounds (10 μg/mL) were dissolved in a MeOH:ACN mixture (8:2, *v*/*v*) and injected with a volume of 3 µL into the ESI system (positive ion mode) using an auto-sampler. The flow rate was 0.2 mL/min. The ESI conditions were set up as follows: ion source and capillary voltages were 4.5 kV and 50 V, respectively. Tube lens offset was 80 V. Capillary temperature was 330 °C. Gas carrier was nitrogen, and the sheath and aux flow rates were 50 arb and 10 arb, respectively. The mass spectra were recorded at 60,000 resolution with a scan range of 100–2000 *m*/*z*. To identify the MS/MS spectra, the PubChem online database (National Center for Biotechnology Information, U.S. National Library of Medicine, Bethesda, MD, USA) and literature were used as references.

### 2.4. Preparation for Germination

The germination process was generated following the method described by Cáceres et al. [29], with several modifications. Nine treatments were applied during the germination stage. The experimental conditions were as follows: soaking time of 36 h, temperature of 30 °C, and different salt (NaCl) concentrations, including 0, 75, and 150 mM. All treatments are presented in Table 1. Germination was conducted for 3, 4, and 5 days in darkness for all treatments. First, 100 g of brown rice was individually measured for nine plastic pots. The rice was soaked in 0.1% NaOCl at a ratio of 1:2 (*w*/*v*) for 30 min to remove or eliminate surface bacteria and fungi without damaging the internal organs [29]. Furthermore, it was also washed five times with clean tap water and dried for 5 min to remove residual NaClO. Then, 75 mM and 150 mM aqueous solutions of NaCl were prepared with distilled water for different treatments. All rice samples were soaked with salinity solution (grain:solution ratio, 1:2 *w*/*v*) and kept in an incubator for various periods at 30 °C (Table 1). Following the soaking period, the seeds were washed with distilled water to remove salinity. All the trays containing brown rice seeds were placed in an incubator at 30 °C for 3, 4, and 5 days in the dark for germination (Table 1). Relative humidity was around 65% in a closed system. The seeds were washed every four hours with distilled water to avoid bacterial and fungal invasions.

### 2.5. Extracted Phytochemicals from GBR

After germination, GBR was washed twice with distilled water and drained for 5 min. The samples were then dried in an oven for 7 days at 40 °C. For extraction, 50 g of GBR powder were saturated in 80% methanol for one week with two replications at room temperature. The extractions were then filtered after centrifugation (10,000 rpm) for 10 min at 4 °C. Subsequently, all methanolic extracts were evaporated at 50 °C to obtain methanol crude extract. Finally, the crude extracts were dissolved in methanol to achieve stock solutions of 20 mg/mL for further experiments.

### 2.6. Total Phenolic Content (TPC) and Total Flavonoid Content (TFC) in GBR Extracts

Total phenolic content (TPC) of the GBR extracts was quantified based on the Folin–Ciocalteu method described by Mohammadabadi et al. [30] with several modifications. Briefly, a mixture of GBR sample, 10% Folin–Ciocalteu’s reagent, and 7.5% Na_2_CO_3_ with volumes of 20, 100, and 80 µL, respectively, was generated and incubated for 30 min at 25 °C in darkness. The results were scanned at 765 nm. Total flavonoid content (TFC) was quantified following the aluminum chloride colorimetric method described in the research of Bueno-Costa et al. [31]. Concretely, a total volume of 100 µL of mixture (1:1, *v*/*v*) comprising GBR sample and 2% AlCl_3_ was incubated for 15 min at 25 °C in darkness. The absorbance was measured at 430 nm. The calibration curves of TPC (0.0052x + 0.0645, *r*^2^ = 0.9969) and TFC (0.009x + 0.0644; *r*^2^ = 0.9998) established by applying the standards of gallic acid and rutin with concentrations ranging from 6.25 to 100 μg/mL were employed for the estimation of TPC and TFC in GBR samples. Of which, TPC and TFC were indicated in milligrams of gallic acid equivalent (GAE) and rutin equivalent (RE), respectively, per one gram of sample dry weight (DW).

### 2.7. Antioxidant Activities of GBR

Radical scavenging activities of GBR extracts were determined via 2,2-diphenyl-1-1-picrylhydrazyl (DPPH) and 2,2′-azino-bis(3-ethylbenzthiazoline-6-sulfonic acid) (ABTS) assays based on the methods presented by Anh et al. [32]. For the DPPH assay, a mixture of GBR sample, DPPH working solution (0.5 mM), and acetate buffer (0.1 mM, pH 5.5) with volumes of 80, 40, and 80 µL, respectively, was incubated for 20 min at 25 °C in darkness. In the ABTS assay, 200 µL of a combination (1:9, *v*/*v*) of GBR sample and ABTS working solution was incubated for 30 min at 25 °C in darkness. The radical scavenging activities (%) were determined as the reduced absorbance at 517 and 734 for DPPH and ABTS assays, respectively, compared to the control (MeOH).

Radical scavenging activity (%) = (A_c_ − (A_s_ − A_b_)/A_c_) × 100
(1)

where A_c_ is the absorbance of the control (MeOH), A_s_ is the absorbance of the sample, and A_b_ is the absorbance of the blank (without radical solution).

### 2.8. Identification and Quantification of Momilactones A (MA) and B (MB) in GBR by Ultra-Performance Liquid Chromatography–Electrospray Ionization-Mass Spectrometry (UPLC–ESI-MS)

MA and MB in GBR samples were identified and quantified by ultra-performance liquid chromatography–electrospray ionization-mass spectrometry (UPLC–ESI-MS). In particular, the UPLC–ESI-MS system consisted of a mass spectrometer (LTQ Orbitrap XL, Thermo Fisher Scientific, Waltham, MA, USA) and an electrospray ionization (ESI) source. A volume of 3.0 μL of GBR sample (in MeOH) was injected by an autosampler (Vanquish autosampler, Thermo Fisher Scientific, Waltham, MA, USA) into a column (1.7 μm, 50 × 2.1 mm i.d.) (Acquity UPLC^®^ BEH C18, Waters Cooperation, Milford, MA, USA) at 25 °C. A mobile phase gradient was applied, of which solvents A and B were 0.1% trifluoroacetic acid in water and 0.1% trifluoroacetic acid in acetonitrile, respectively. The gradient program was established following the same procedure published by Anh et al. [33]. MS analysis was conducted with a positive Fourier transform mass spectrometer (FTMS) mode with 60,000 resolution and 100–1000 *m*/*z* of scan range. By using various MA and MB standard concentrations (0.5, 1, 5, and 10 µg/mL), the calibration curves for MA and MB were created. Using standard curves, MA and MB quantities were determined by applying the MA and MB peak areas detected in each sample.

### 2.9. Identification and Quantification of Tricin, ρ-Coumaric Acid, Ferulic Acid, Cinnamic Acid, and Salicylic Acid by High-Performance Liquid Chromatography (HPLC)

The presence and quantification of tricin, *ρ*-coumaric acid, ferulic acid, cinnamic acid, and salicylic acid by high-performance liquid chromatography (HPLC) analyses were compared with the standards attained by the method presented by Anh et al. [33]. In brief, the HPLC system consisted of a pump (PU-4180 RHPLC, Jasco, Tokyo, Japan), a controller (LC-Net II/ADC, Jasco, Japan), and a detector (UV-4075 UV/VIS, Jasco, Tokyo, Japan). A column (130 Å, 5 µm, 2.1 × 100 mm) (XBridge BEH Shield RP18, Waters Cooperation, Milford, MA, USA) was used as a stationary phase. Solvent A (0.1% formic acid in water) and solvent B (acetonitrile) were applied as mobile phases, which were fixed in the same gradient program reported by Anh et al. [33]. Each operation was continued for 35 min at room temperature. Every sample was identified by a corresponding peak scanned at 350 nm for tricin and 280 nm for *ρ*-coumaric acid, ferulic acid, cinnamic acid, and salicylic acid. The peak area was used to quantify these compounds.

### 2.10. Statistical Analysis

All experiments were conducted in a completely randomized design (CRD) with three replications. The analyses were performed using Minitab software (Minitab 16.2.3, Minitab Inc., State College, PA, USA) through one-way and two-way ANOVA. The outcomes were presented as means ± standard deviations (SD) (*n* = 3). The same software was used for Pearson’s correlation coefficients among different parameters.

## 3. Results and Discussion

### 3.1. Confirmation of Isolated Tricin, ρ-Coumaric Acid, and Momilactones A (MA) and B (MB)

Isolated tricin, *ρ*-coumaric acid, and momilactones A (MA) and B (MB) were identified and confirmed using electrospray ionization-mass spectrometry (ESI-MS) and ^1^H- and ^13^C-nuclear magnetic resonance (NMR) methods. The mass spectra of these compounds are shown in Figure 1.

Tricin: ESI-MS (*m*/*z*): 331.08139 [M + H]^+^ (C_17_H_15_O_7_) (Figure 1). The mass spectrum of tricin was compared with that in published data by Quan et al. [13]. The ^1^H NMR (600 MHz, DMSO-d_6_) δ 12.90 (1H, s, 5-OH), 10.84 (d, *J* = 124.2, 7-OH), 9.26 (s, 1H, 4-OH), 7.26 (s, H-60 and H-20), 6.92 (s, H-3), 6.49 (d, *J* = 2.1, H-8), 6.13 (d, *J* = 2.1, H-6), 3.82 (s, 2OCH_3_), 3.26 (s, 220H), 2.43 (dt, *J* = 3.6 and 1.8, 171H) (Appendix A). The ^13^C NMR (151 MHz, DMSO-d6) δ 182.35 (C-4), 164.66 (C-2), 164.19 (C-7), 161.94 (C-5), 157.87 (C-9), 148.72 (C-30 and C-50), 140.38 (C-40), 56.90 (2OCH_3_), 40.05 (dp, *J* = 42.0, 21.0 Hz) (Appendix A). The NMR results of tricin are matched with reference data from a previous study [34].

*ρ*-Coumaric acid: ESI-MS (*m*/*z*): 165.05424 [M + H]^+^ (C_9_H_9_O_3_) (Figure 1). The ^1^H NMR (400 MHz, DMSO-d_6_) δ 7.51 (dd, *J* = 16.0 and 8.6, H-7, H-2, H-6), 6.79 (d, *J* = 8.6, H-3, H-5), and 6.29 (d, *J* = 16.0, H-8) (Appendix A). The ^13^C NMR (101 MHz, DMSO-d_6_) δ 168.42 (COOH), 160.05 (C-4), 144.64 (C-7), 130.53 (C-2, C-6), 125.75 (C-1), 116.22 (C-3, C-5), and 115.80 (C-8) (Appendix A). The NMR and ESI-MS results are entirely similar to those in the literature [35].

MA: ESI-MS (*m*/*z*): 315.19470 [M + H]^+^ (C_20_H_27_O_3_) (Figure 1). The mass spectrum of MA was confirmed based on reference data reported by Quan et al. [24]. The ^1^H-NMR (600 MHz, CDCl_3_) δ 5.87 (s, 1H), 5.86 (s, 1H), 5.85 (s, 1H), 5.83 (d, J = 17.0, 11.0, H-15), 5.71 (d, *J* = 5.0, H-7), 5.00 (d, *J* = 1.0, 1H), 4.97 (d, *J* = 1.0, 1H), 4.95 (d, *J* = 1.0, 1H), 4.93 (d, *J* = 1.0, 1H), 4.84 (t, *J* = 5.1, H-6), 4.10–4.06 (m, 1H), 3.97 (s, 1H), 3.95 (s, 1H), 3.30 (s, 1H), 3.28 (s, 1H), 3.21 (s, 1H), 3.18 (s, 1H), 2.67–2.56 (m, H-2), 2.32 (d, *J* = 5.1, H-5), 2.21 (d, *J* = 12.0, 2H-14), 2.08–2.04 (m, 1H), 1.93–1.86 (m, 1H), 1.79 (dd, *J* = 12.9, 3.9, 1H), 1.77–1.72 (m, H-9, H-11α), 1.63–1.55 (m, H2-1β, H2-12), 1.53 (s, H-18), 1.00 (s, H-20), 0.89 (s, H-17) (Appendix A). The ^13^C NMR (151 MHz, CDCl_3_) δ 205.24 (C-3), 174.37 (C-19), 149.03 (C-8), 148.10 (C-15), 114.12 (C-7), 110.25 (C-16), 73.23 (C-6), 53.64 (C-4), 50.26 (C-9), 47.60 (C-14), 46.54 (C-5), 40.20 (C-13), 37.31 (C-12), 34.95 (C-1), 32.53 (C-10), 31.29 (C-2), 24.06 (C-11), 22.03 (C-20), 21.87 (C-17), 21.54 (C-18) (Appendix A). The NMR spectrum is matched with published data in the report of Quan et al. [23].

MB: ESI-MS (*m*/*z*): 331.19006 [M + H]^+^ (C_20_H_27_O_4_) (Figure 1). The obtained results were confirmed by comparing them with those in the preceding report [24]. The ^1^H NMR (600 MHz, CDCl_3_) δ 5.82 (dd, *J* = 17.5, 10.7, H-15), 5.69 (d, *J* = 4.8, H-7), 4.98–4.92 (m, 1H), 4.13 (s, 1H), 4.08 (dd, J = 9.2, 3.4, 1H), 3.58 (dd, J = 9.2, 2.1, 1H), 2.20 (dd, J = 6.8, 2.0, H-5), 2.14–2.07 (m, H-2, H-14), 2.04–1.98 (m, 1H), 1.75–1.64 (m, H-9, H-11α), 1.59–1.57 (m, 1H), 1.57–1.51 (m, H-1β, H-12), 1.50 (d, *J* = 4.2, 1H), 1.48–1.43 (m, 1H), 1.41 (s, H-18), 1.26–1.19 (m, 1H), 0.87 (s, H-17) (Appendix A). The ^13^C NMR (151 MHz, CDCl_3_) δ 180.53 (C-19), 148.91 (C-15), 146.76 (C-8), 114.09 (C-7), 110.30 (C-16), 96.67 (C-3), 73.81 (C-6), 72.79 (C-20), 50.41 (C-4), 47.49 (C-14), 44.76 (C-9), 43.05 (C-5), 40.06 (C-13), 37.29 (C-12), 30.81 (C-10), 28.89 (C-1), 26.51 (C-2), 24.86 (C-11), 21.94 (C-17), and 19.06 (C-18) (Appendix A). NMR data were compared to published results by Quan et al. [23].

### 3.2. Phenolic and Momilactone Contents in GBR

#### 3.2.1. Total Phenolic (TPC) and Flavonoid (TFC) Contents

In essence, the chemical composition of natural products determines their biological activity [32]. Additionally, methods for the identification and quantification of natural compounds have been rapidly advancing [36]. Therefore, exploring the phytochemical profiles of targeted products is necessary for studies concerning their potential bioactivity. In our research, the initial assessment focused on the compound groups of phenolics and flavonoids, which may contribute to the pharmaceutical and medicinal properties (e.g., antioxidant, antibacterial, anticancer, cardioprotective, immune system-promoting and anti-inflammatory, and skin-protective effects) of the targeted products [37]. The total phenolic (TPC) and flavonoid (TFC) contents of GBR are shown in Figure 2. There were significant differences in TPCs and TFCs among different treatments. Particularly, the highest TPCs were found in treatments B2 and C2 (14.50 and 14.36 mg GAE/g DW, respectively). Whereas the lowest TPC was observed in A2 (6.17 mg GAE/g DW). On the other hand, the highest TFC (11.06 mg RE/g DW) was detected in B2, while the lowest TFC (2.54 mg RE/g DW) was found in A2. Previous studies indicated that TPC and TFC increased in rice seedlings under salinity effects [38,39,40], which might be due to the upregulation of genes encoding the major biosynthetic enzymes (e.g., phenylalanine ammonia lyase and chalcone synthase) in plant responses to biotic stresses [33,41,42]. In this study, TPC and TFC were remarkably stimulated in treatments with 75 mM salinity. However, they were remarkably decreased when increasing to an extreme salt level of 150 mM. Our findings suggest that a moderate salinity level of 75 mM and 4-day germination are the most appropriate conditions for proliferating TPC and TFC in GBR.

Considering TPC analysis, the Folin–Ciocalteu method is widely used because it provides a quick and cost-effective estimation, allowing for comparative analyses between samples [43]. However, the method reveals limitations in specifically quantifying TPC due to its reactivity with other components such as amino acids, peptides, and reducing sugars [43]. For a more accurate determination of TPC, an improved method such as solid-phase extraction using the Sep-Pak C18 column cartridges is required to purify the extract and eliminate unwanted components [30]. Though the Folin–Ciocalteu method serves as a valuable screening tool to assess the relative phenolic content in various samples, it may not provide precise quantification of individual phenolic compounds [43]. Therefore, in further investigation, we employed HPLC analysis to determine the profiles of specific phenolics found in GBR in this study.

#### 3.2.2. Contents of Tricin, *ρ*-Coumaric, Ferulic, Cinnamic, and Salicylic Acids in GBR

Tricin, *ρ*-coumaric acid, ferulic acid, cinnamic acid, and salicylic acid play multifunctional roles benefiting human health, including antioxidants, anticancer, and anti-chronic diseases [44]. In the present study, these phenolic compounds were identified (Appendix A) and quantified (Table 2). Accordingly, tricin, *ρ*-coumaric acid, ferulic acid, salicylic acid, and cinnamic acid were found in increasing quantities in B2 treatment (75 mM salinity and 4-day germination), which accounted for 107.63, 93.77, 139.03, 46.05, and 596.26 µg/g DW, respectively. The elevated contents of *ρ*-coumaric, salicylic, and ferulic acids are in line with those in rice seedlings subjected to salinity (100 mM) in a previous study [38]. Meanwhile, cinnamic acid and tricin amounts decreased [38], which does not align with our findings. This might be due to the differences in genetic diversity among tested rice varieties. In fact, different rice varieties (tolerant and susceptible cultivars) exhibit dissimilar mechanisms in phenolic accumulation to cope with stress conditions [33,38,39,40]. On the other hand, the present study revealed a notable decrease in the quantities of these phenolics in GBR when exposed to extreme levels of salinity (150 mM). This finding implies that the most optimal conditions for the proliferation of bioactive phenolics in GBR are a salinity of 75 mM maintained for 4 days during the germination process.

#### 3.2.3. Contents of Momilactones A (MA) and B (MB) in GBR

Momilactones A (MA) and B (MB) have been known as valuable bioactive compounds from rice with various health-related benefits, including antioxidant [13], anticancer (leukemia [20], lymphoma [21], and colon cancer [22]), anti-diabetes [23,24], anti-obesity [24], and anti-skin aging activities [13,45]. Recently, based on an improved technique for sample preparation and quantification, momilactones can also be detected with high detection sensitivity in different rice plant parts (e.g., leaves, roots, husks, etc.) [23,45,46]. However, their exploitation from rice sources has still faced many limitations due to the lack of commercial availability and difficulties in the isolation process [23,45]. A few reports about momilactone isolation and purification have been published, and in those studies, a minor amount of momilactones can be isolated from rice sources [45,46,47,48]. Additionally, the published artificial syntheses of MA were also challenging since they included multiple complicated steps, required high costs, resulted in low yields (40–50%), and were environmentally unfriendly [49]. On the other hand, the synthetic methods of MB have never been reported. In fact, due to their limited availability, research on momilactones has been relatively scarce and underdeveloped during the last half-century. Recently, only two studies focused on optimizing the extraction conditions of MA and MB from rice husks [48,50], while no research has been conducted to enhance the momilactone contents of rice grains to increase their consumption value. Therefore, this study investigated, for the first time, the effects of different conditions (salinity and germination periods) on the accumulation of MA and MB in GBR. In Appendix A, the presence of MA and MB in GBR is confirmed by comparing their retention times and mass spectra with those of the standards. Numerous studies indicated that the antioxidant, anti-diabetic, and anticancer potentials of MB were greater than those of MA [23,51,52]. However, Chung et al. [53] announced that the endogenous quantity of MA was generally higher than that of MB in different 99 rice varieties. In contrast to previous reports, our findings demonstrate that Koshihikari GBR exhibited a greater amount of MB than MA in all treatments (Table 2). Significantly, the highest accumulation of MA (18.94 µg/g DW) and MB (41.00 µg/g DW) was recorded in the B2 treatment. The amounts of MA and MB in GBR under B2 were significantly higher than those of preceding studies, in which MA and MB quantities ranged from 2.07 to 16.44 µg/g DW and 1.06 to 12.73 µg/g DW, respectively [13,23,24]. The increased accumulation of MA and MB in GBR in B2 treatment may be caused by the elevated expression of related genes to momilactone biosynthesis, including OsCPS4, OsKSL4, CYP99A3, OsMAS, and OsMAS2 [33]. However, at a strong salinity level of 150 mM, both MA and MB accumulations in GBR were significantly reduced compared to non-salinity treatment. The results suggest that a moderate salt concentration of 75 mM and 4-day germination are the most ideal conditions to stimulate MA and MB contents, which may contribute to the pharmaceutical and medicinal values of GBR.

### 3.3. Antioxidant Activity of GBR by the DPPH and ABTS Radical Scavenging Assays

In humans, oxidative stress is closely associated with inflammation, which is considered a key physiological process in the development of various chronic diseases such as diabetes, aging, and cancer [54,55]. Specifically, inflammation can worsen oxidative stress and vice versa [54,55]. Numerous experimental findings have demonstrated the presence and impact of oxidative stress in several chronic diseases, which result in higher rates of morbidity and mortality [54,55]. Based on that, assessing the antioxidant properties of the samples is an essential step in our study. According to Figure 3, changes in salinity levels and germination periods might promote the antioxidation activity of GBR. Among all treatments, B2 showed the highest antiradical activities against DPPH and ABTS (IC_50_ = 1.58 and 1.78 mg/mL, respectively) compared to others. Several studies have consistently demonstrated that different germination conditions and salt stress enhanced the antioxidant capacity of GBR [6,56,57]. This result may be due to the upregulation of relevant enzymes such as superoxide dismutase, catalase, glutathione peroxidase, and ascorbate peroxidase in rice under abiotic stresses [58]. Therefore, salinity effects could potentially contribute to enhancing the antioxidant capacities of GBR in this research. However, the salt concentration should be carefully considered since extreme levels may lead to reduced antioxidant capacities of GBR [59]. In agreement with Falcinelli et al. [59], we indicate that the antioxidant activity of GBR significantly decreased at a high NaCl concentration of 150 mM. Meanwhile, a moderate salinity level of 75 mM may be the most effective condition to elevate the antioxidant activity of GBR.

### 3.4. Correlation between Antioxidant Activities and Phytochemicals of GBR

Pearson’s correlation coefficients between antioxidant activities and phytochemicals are displayed in Table 3. Accordingly, a concomitant accumulation of MA, MB, tricin, *ρ*-coumaric, ferulic, cinnamic, and salicylic acids is recorded, which was strongly correlated with the antioxidant activities of GBR. Previous studies have extensively reported on the roles of tricin, *ρ*-coumaric, ferulic, cinnamic, and salicylic acids in antioxidant abilities [60]. Particularly, these compounds can detoxify free radicals by donating hydrogen ions, thus strengthening antioxidant capacities [60]. Conversely, while the antioxidant activities of MA and MB have been mentioned in several publications [13,52], their underlying mechanisms remain unclear. In another consideration, Anh et al. [33] hypothesized that MA and MB might not directly contribute to the antioxidant responses of rice against adverse stresses, but they might play a role in signaling the production of antioxidant compounds such as phenolics [33]. Based on that, in this study, the upregulated contents of MA and MB in GBR under moderate salinity (75 mM NaCl) and 4-day germination might lead to the proliferation of tricin, *ρ*-coumaric, ferulic, cinnamic, and salicylic acids, thereby increasing the antioxidant capacity of GBR, which requires further validation.

Additionally, due to the well-established correlation between oxidative stress and chronic diseases [54,55], extensive research has been undertaken to explore the use of antioxidant substances for the treatment of such disorders [61]. However, the failures have been documented through clinical evaluations, which might be attributed to the single use of antioxidant agents to target specific diseases [61]. Furthermore, interactions among compounds may hold greater significance than individual ones, leading to enhanced therapeutic efficiency [13,32,62]. Therefore, the simultaneous proliferation of bioactive compounds and antioxidant activity of GBR in this study might potentially lead to a synergistic effect that benefits human health. Our findings may support the promotion of rice consumption values as well as the development of pharmaceuticals, functional foods and supplements. For example, GBR treated with B2 (75 mM salinity for 4 days) can be applied to produce a fermented functional beverage known as kombucha, which has become increasingly popular because of its health benefits [63]. On the other hand, considering the impacts of human digestion on targeted products, their bioaccessibility and bioavailability during the digestion process should be thoroughly examined in future studies [64].

## 4. Conclusions

This research, for the first time, has identified an optimized treatment (B2: 75 mM NaCl and 4-day germination) that significantly advanced the accumulation of valuable bioactive compounds, including phenolics and momilactones A (MA) and B (MB), in germinated brown rice (GBR, Koshihikari var.). In particular, GBR treated by B2 contained the highest amounts of total phenolics and total flavonoids. Moreover, the isolated bioactive compounds were identified and confirmed by electrospray ionization-mass spectrometry (ESI-MS) and nuclear magnetic resonance (NMR) spectroscopy (^1^H and ^13^C). Additionally, the quantification results indicated that GBR under B2 treatment accumulated the greatest quantities of MA, MB, tricin, *ρ*-coumaric acid, ferulic acid, cinnamic acid, and salicylic acid. The B2 treatment also significantly enhanced the antioxidant activities of GBR, as demonstrated in the antiradical assays (DPPH and ABTS). In the context that BR has been less favored, leading to improper utilization or waste of this source, the outcomes of our study hold promising prospects for enhancing the nutritional value of BR and fostering the advancement of rice-derived products that contribute to human well-being. Consequently, this research serves to incentivize the consumption of BR by highlighting its intrinsic value and potential benefits for human health. Furthermore, the present findings are expected to contribute to the attainment of the Sustainable Development Goals (SDGs) by promoting the overall welfare of individuals, eradicating poverty, and ensuring global food security, particularly in countries reliant on rice cultivation.

## Figures and Tables

**Figure 1 foods-12-02501-f001:**
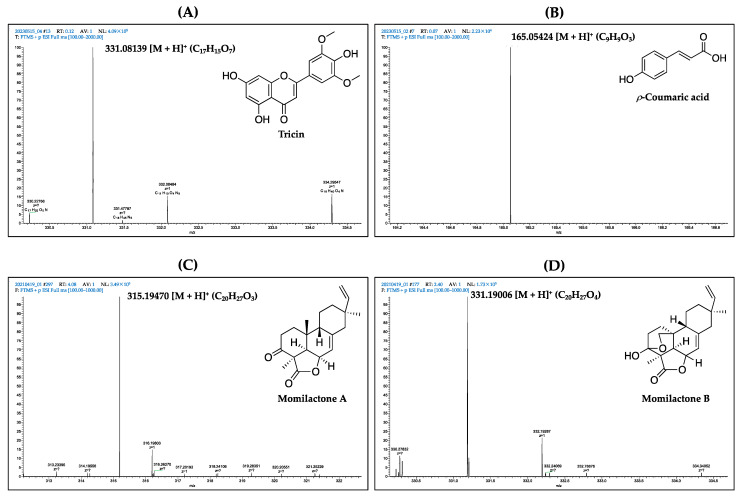
Mass spectra of isolated (**A**) tricin, (**B**) *ρ*-coumaric acid, (**C**) momilactone A (MA), and (**D**) momilactone B (MB) in this study by ESI-MS.

**Figure 2 foods-12-02501-f002:**
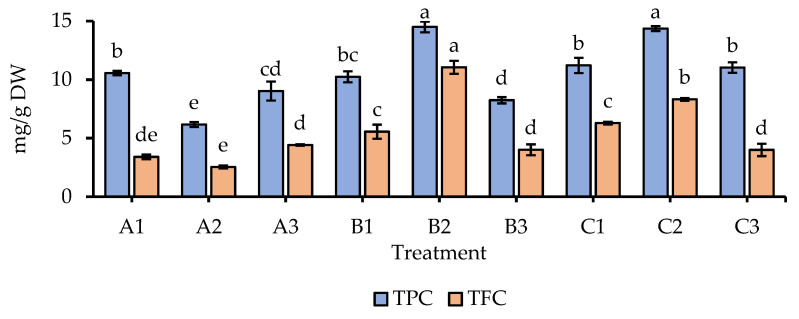
Total phenolic (TPC) and total flavonoid (TFC) contents of GBR extracts. TPC and TFC outcomes are expressed as mg gallic acid equivalent per g dry weight (mg GAE/g DW) and mg rutin equivalent per g dry weight (mg RE/g DW), respectively. Whiskers enclosed in a column express the standard deviation (SD). Different letters attached to a column (same color) indicate significant differences at *p* < 0.05. A1: 0 mM NaCl and 3-day germination; A2: 75 mM NaCl and 3-day germination; A3: 150 mM NaCl and 3-day germination; B1: 0 mM NaCl and 4-day germination; B2: 75 mM NaCl and 4-day germination; B3: 150 mM NaCl and 4-day germination; C1: 0 mM NaCl and 5-day germination; C2: 75 mM NaCl and 5-day germination; C3: 150 mM NaCl and 5-day germination.

**Figure 3 foods-12-02501-f003:**
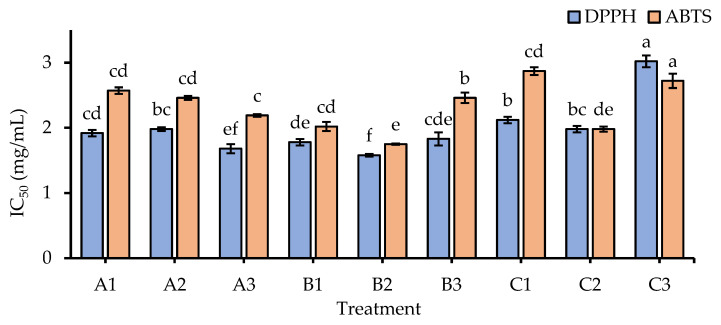
Antioxidant activities of GBR extracts. IC_50_ is the required concentration (mg/mL) for scavenging 50% of radicals. Whiskers enclosed in a column express the standard deviation (SD). Different letters attached to a column (same color) indicate significant differences at *p* < 0.05. DPPH: 2,2-diphenyl-1-picrylhydrazyl assay; ABTS: 2,2′-azino-bis(3-ethylbenzothiazoline-6-sulfonic acid assay; A1: 0 mM NaCl and 3-day germination; A2: 75 mM NaCl and 3-day germination; A3: 150 mM NaCl and 3-day germination; B1: 0 mM NaCl and 4-day germination; B2: 75 mM NaCl and 4-day germination; B3: 150 mM NaCl and 4-day germination; C1: 0 mM NaCl and 5-day germination; C2: 75 mM NaCl and 5-day germination; C3: 150 mM NaCl and 5-day germination.

**Table 1 foods-12-02501-t001:** Description of different treatments.

Treatments Code	NaCl Concentration (mM)	Germination Time (Day)
A1	0	3
A2	75	
A3	150	
B1	0	4
B2	75	
B3	150	
C1	0	5
C2	75	
C3	150	

**Table 2 foods-12-02501-t002:** Quantities of momilactones, tricin, *ρ*-coumaric acid, ferulic acid, cinnamic acid, and salicylic acid (µg/g DW) in GBR.

Treatment Code	MA	MB	Tricin	*ρ*-Coumaric Acid	Ferulic Acid	Cinnamic Acid	Salicylic Acid
A1	7.33 ± 0.39 ^c^	18.68 ± 0.89 ^c^	44.43 ± 8.92 ^cd^	46.43 ± 3.37 ^de^	64.92 ± 3.34 ^bc^	28.16 ± 0.64 ^b^	290.27 ± 68.05 ^bc^
A2	2.92 ± 0.06 ^ef^	9.30 ± 0.09 ^e^	41.12 ± 5.57 ^cde^	39.88 ± 0.63 ^ef^	53.23 ± 1.88 ^c^	21.86 ± 1.09 ^c^	349.04 ± 83.3 ^b^
A3	1.93 ± 0.09 ^f^	7.27 ± 0.12 ^f^	29.55 ± 2.04 ^ef^	36.29 ± 1.66 ^f^	49.05 ± 2.72 ^c^	22.58 ± 1.10 ^c^	194.16 ± 14.77 ^cd^
B1	5.68 ± 1.38 ^cd^	18.88 ± 0.57 ^c^	33.93 ± 1.38 ^def^	61.77 ± 1.96 ^b^	57.22 ± 7.63 ^bc^	11.82 ± 0.82 ^d^	88.49 ± 22.89 ^de^
B2	18.94 ± 0.47 ^a^	41.00 ± 0.51 ^a^	107.63 ± 6.75 ^a^	93.77 ± 4.35 ^a^	139.03 ± 5.16 ^a^	46.05 ± 0.88 ^a^	596.26 ± 1.14 ^a^
B3	4.19 ± 0.03 ^de^	12.70 ± 0.75 ^d^	25.51 ± 0.94 ^f^	44.99 ± 1.44 ^def^	61.98 ± 2.36 ^bc^	22.01 ± 0.52 ^c^	52.86 ± 3.2 ^e^
C1	4.90 ± 0.17 ^de^	11.97 ± 0.05 ^d^	49.54 ± 0.34 ^c^	44.48 ± 2.90 ^def^	52.55 ± 4.77 ^c^	-	-
C2	10.17 ± 0.49 ^b^	24.79 ± 0.55 ^b^	65.13 ± 3.06 ^b^	59.95 ± 5.51 ^bc^	76.55 ± 9.07 ^bc^	-	-
C3	1.70 ± 0.01^f^	7.20 ± 0.29 ^f^	31.67 ± 0.59 ^ef^	51.52 ± 3.19 ^cd^	60.81 ± 5.97 ^bc^	-	-
ANOVA							
Period	***	***	***	***	***	***	***
Treatment	***	***	***	***	***	***	***
Period × Treatment	***	***	***	***	***	***	***

Data are expressed as means ± SD (standard deviation). Different superscript letters (^a,b,c,d,e,f^) in a column indicate significant differences at *p* < 0.05; *** denotes a significant difference at *p* < 0.001. MA: momilactone A; MB: momilactone B; DW: dry weight; -: not detected; A1: 0 mM NaCl and 3-day germination; A2: 75 mM NaCl and 3-day germination; A3: 150 mM NaCl and 3-day germination; B1: 0 mM NaCl and 4-day germination; B2: 75 mM NaCl and 4-day germination; B3: 150 mM NaCl and 4-day germination; C1: 0 mM NaCl and 5-day germination; C2: 75 mM NaCl and 5-day germination; C3: 150 mM NaCl and 5-day germination.

**Table 3 foods-12-02501-t003:** Pearson’s correlation coefficients between phytochemicals and antioxidant activities of GBR.

	MA	MB	*ρ*-Cou	Tri	Fer	Sal	Cin	DPPH	ABTS	TFC
**MB**	0.984 ***									
***ρ*-Cou**	0.888 ***	0.915 ***								
**Tri**	0.940 ***	0.898 ***	0.837 ***							
**Fer**	0.908 ***	0.901 ***	0.905 ***	0.886 ***						
**Sal**	0.610 **	0.572 **	0.497 *	0.613 **	0.653 **					
**Cin**	0.548 **	0.526 **	0.419 *	0.458 *	0.594 **	0.900 ***				
**DPPH**	0.510 **	0.540 **	0.355 *	0.397 *	0.431 *	0.577 **	0.722 **			
**ABTS**	0.053	0.043	0.063	0.003	0.259	0.18	0.347 *	0.041		
**TFC**	0.860 ***	0.858 ***	0.851 ***	0.864 ***	0.809 ***	0.304 *	0.216	0.410 *	−0.07	
**TPC**	0.743 **	0.744 **	0.733 **	0.727 **	0.670 **	0.068	−0.029	0.09	−0.113	0.861 ***

*, **, and *** indicate significances at *p* < 0.05, 0.01, and 0.001, respectively; MA: momilactone A; MB: momilactone B; *ρ*-Cou: *ρ*-coumaric acid; Tri: tricin; Fer: ferulic acid; Sal: salicylic acid; Cin: cinnamic acid; TPC: total phenolic content; TFC: total flavonoid content. ABTS: 2,2′-azino-bis-(3-ethylbenzothiazoline-6-sulfonic acid) assay; DPPH: 2,2-diphenyl-1-picrylhydrazyl assay.

## Data Availability

The data used to support the findings of this study can be made available by the corresponding author upon request.

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
