# Peer review of "Salinity Treatments Promote the Accumulations of Momilactones and Phenolic Compounds in Germinated Brown Rice"

_foods, 2023, doi:10.3390/foods12132501_

Round 1
Reviewer 1 Report
The present manuscript is, in general, well written and discussed. Must improve the conclusion section and update the references. With regard to improvements in the methodology section, as well as in general, the article presents numerous self-citations, which should be changed, as well as more current references.
The conclusion section should be improved, presenting in more detail the results found in the study.
Captions and configuration of tables and figures should be improved.
Please make the requested changes.
The present manuscript is, in general, well written and discussed. Small corrections should be made (as in line 33 "procures" and line 440 "to augment") and also improve figures and tables captions.
Author Response
Dear Respected Reviewer,
We sincerely thank you for your valuable comments and suggestions. The manuscript has been extensively revised following your advice. Please check the attachment, which includes our detailed responses to your feedback.
Sincerely thanks,
Tran Dang Xuan
Corresponding author
On behalf of all authors

Reviewer 2 Report
Abstract: The author needs to restructure the abstract. The abstract should begin with a brief but precise statement of the problem or issue, followed by the research method, the major findings, and the conclusions reached
Introduction/Method
• Not enough depth in background and literature information on research study
• Part of English writing needs to be recheck.
• Suggest the author can include some physical characteristics of rice and rice husks variety koshhkari. The authors did not describe in detail this topic in the introduction. It is good information for the reader.
• The authors did not describe problem statement in detail in the introduction.
- The author also not describe the aim of the study in the introduction
• Selection of rice husk koshhvari also not describe and not clear description on the sample preparation.
• Also not clear why rice husk koshhvari was chosen
• Information for instruments used must be given consistently and completely. example: name of equipment (Model, brand, and country).
Result and discussions
• The resulting found was well presented; however, the lack of critical discussion was made in discussions. Proven result of the past literature and other related points should/must have at every argument.
• All the Tables and the figure require proper legends and footnotes, explaining all the symbols.
• There are few typing errors. Please recheck
Add Conclusion
• The author should highlight the best conclusion and relate with the aim of study. The conclusion should include:
==> Discus in brief about type of sample and related with the finding results
==> Justify the reason for the importance of the results
moderate. need to proofread
Author Response

(The authors gave the same response as above.)

Reviewer 3 Report
Dear Authors,
This is promising study which is dealing with brown rice seeds in order to highlight phenolic compounds in wine, and to improve nutritional potential of brown rice .
There is more words in abstract than in instructions. Correct it.
Did you treat rice crops with any pesticide?
In the line 67 highlight that other natural products such as wine are good source of phenolic acids . Kindly consider to cite Maced. J. Chem. Chem. Eng., 39, No. 2, (2020) 185–196.
What was the waiting period after treatment of potatoes with preparations for protection against pests?
What are climate conditions in this part of Japan. especially in December when you conducted study? Highlight more about it.
How findings from your study can be applied in practice?
Wish you all the best in future work.
Minor editing of English language required.
Author Response
Dear Respected Reviewer,
Thank you very much for your comments. We would like to respond to your feedback as in the attachment. Please kindly check the attached file.
Sincerely thanks,
Tran Dang Xuan
Corresponding author
On behalf of all authors

Reviewer 4 Report
Hi dear Editorial board and the respected authors
This article "Variations of Momilactones and Phenolic Compounds in Brown Rice Seeds Germinated under Salinity Treatments” was revised and has a novelty and I recommend it for publication after consideration of the following comments.
Title: If you can rewrite and make it more interesting for readers.
Abstract:
· The type of statistical design and the treatments types used in this research should be mentioned.
· The details of the data of experiments must be better mentioned in the abstract.
· Line 19 for the first time of mention must be express the completely word for example RE and GAE etc.
· Please say the background of momilactones in the first of abstract.
Introduction:
· Line 52: minerals (e.g., vitamin B and total proteins) are you think vitamins and total protein components of minerals? Please correct it scientifically approach.
· Line 77-80: Please include the detail of treatments provided in your study another time.
Materials:
· Please write materials as Company Name (City, Country), especially for chemical analysis assessment which used in the study.
· Line 132-133: “The rice was soaked in 1% sodium hypochlorite at a ratio of 1:2 (w/v) for 30 min to remove or eliminate surface bacteria and fungi without damaging the internal organs” please point to the references and I think this time is very long because in the less times you can be better approach for killing bacteria and have not any side effect on internal components of brown rice. Please have a new reference.
“Results:
· All Tables i.e., Table 1: The alphabetical statistical letters for the means should all be modified such that the greatest number has the letter a and as the numbers go lower, letters b, c etc.
· Line 152-166: Please include the formula of regression standard curve of TPC and TFC
· Line 154: Folin-Ciocalteu method is not accuracy method for TPC assay because the indicator react with the another components e.g. amino acids peptides reducing sugars etc. please express why did you use this method.
· What is different between ABTS and DPPH assay and include it somewhere in introduction section.
· Please cite the “Foods 2022, 11(20), 3263; https://doi.org/10.3390/foods11203263” for analysis and assay of bioactive compounds.
Discussion:
Discussion text must grammar improve and in some cases it is very weak and maybe there is no discussion at all
Conclusions:
Conclusion is very general, try to make it more scientific, comprehensive and concise in detail, especially.
References: It is OK.
The article has many flaws in express and concept of English, it is suggested to be revised in a scientific and native way.

The article has many flaws in express and concept of English, it is suggested to be revised in a scientific and native way.
Author Response

(The authors gave the same response as above.)

Round 2
Reviewer 3 Report
Dear Dr Tran Dang Xuan,
Thank you very much for revised version of your manuscript. It is fine. Wish you all the best in future work.
Best regards,